# Trauma and Mental Health Awareness in Emergency Service Workers: A Qualitative Evaluation of the Behind the Seen Education Workshops

**DOI:** 10.3390/ijerph18094418

**Published:** 2021-04-21

**Authors:** Andrea Fogarty, Zachary Steel, Philip B. Ward, Katherine M. Boydell, Grace McKeon, Simon Rosenbaum

**Affiliations:** 1Black Dog Institute, Randwick, Sydney 2031, Australia; a.fogarty@blackdog.org.au (A.F.); k.boydell@unsw.edu.au (K.M.B.); 2School of Psychiatry, University of New South Wales, Sydney 2052, Australia; z.steel@unsw.edu.au (Z.S.); p.ward@unsw.edu.au (P.B.W.); g.mckeon@unsw.edu.au (G.M.); 3St John of God Health Care, Richmond Hospital, Sydney 2754, Australia; 4Schizophrenia Research Unit, South Western Sydney Local Health District & Ingham Institute of Applied Medical Research, Liverpool 2170, Australia

**Keywords:** mental health, PTSD, first-responders, occupational stress, emergency service workers, qualitative, focus groups

## Abstract

Emergency service workers (ESWs) are at high risk of experiencing poor mental health, including posttraumatic stress disorder (PTSD). Programs led by ex-service organizations may play an unrecognized but critical role in mental health prevention and promotion. Behind the Seen (BTS) is an Australian ex-service organization that runs workshops to raise awareness and facilitate conversations around the mental health of ESWs. The purpose of the study is to conduct a qualitative evaluation of workshop participants’ experiences, to understand the acceptability and perceived usefulness over the immediate- (within 1 month), intermediate- (6 months) and longer-terms (12 months). Participants (*n* = 59 ESWs) were recruited using purposive sampling across five fire and rescue services in metropolitan, regional, and rural locations. Focus groups methodology was used for data collection and data were analyzed using iterative categorization techniques. Participants reported (i) a high perceived need for education about PTSD, (ii) highly salient aspects of the presentation that made for a positive learning experience, including the importance of the lived experiences of the facilitators in the learning process, (iii) key features of changes to intentions, attitudes, and behavior, and (iv) major aspects of the organizational context that affected the understanding and uptake of the program’s key messages. BTS was perceived as an acceptable means of delivering mental health, PTSD, and help-seeking information to ESWs. The program is a promising candidate for scaling-up and further translation.

## 1. Introduction

Emergency service workers (ESWs), including firefighters, are regularly exposed to trauma as part of their work. The repeated exposure places them at high risk of poor mental health, including posttraumatic stress disorder (PTSD) [1]. There is increasing recognition of the impact of emergency service work [2,3], with poor social support, high occupational stress and maladaptive coping strategies associated with an increased prevalence of mental health problems [2,4,5]. ESWs are also twice as likely to experience suicidal thoughts [6], and one ESW dies by suicide in Australia every four weeks [7]. Alongside occupational exposure to workplace stress and traumatic incidents, significant risk factors for developing PTSD among ESW include previous mental health symptoms related to depression and/or substance use [8], earlier life trauma [9,10], and sustaining injuries or exposure to physical danger during a traumatic incident [11,12]. There are significant barriers that prevent early detection and help-seeking for PTSD among ESWs. Previous research identified an initial lack of knowledge about PTSD among individuals and organizations, which is further compounded by avoidance of trauma-related memories among affected populations, individual beliefs about both the ineffectiveness of treatment and of the need for self-reliance in solving problems, access barriers related to time and money, and a lack of trust in institutionally based services, compounded by stigma and perceived shame regarding help-seeking [13,14]. A key concern for first responders can be the impact of a disclosure of mental health problems on their careers, including the potential to affect their ability to continue working in emergency services [15].

Mental health literacy is often conceptualized as knowledge of and attitudes towards both mental illness and help-seeking [16]. Psychoeducation and mental health literacy interventions are widely acknowledged as crucial for improving knowledge, changing attitudes towards seeking help for mental health problems, and reducing barriers to treatment [17]. One potential strategy for minimizing risk is psychoeducation about PTSD targeting ESWs, in order to facilitate earlier detection and treatment. Psychoeducation about PTSD includes developing an understanding of PTSD, accepting its prevalence among ESW, normalizing the symptoms of PTSD, and adopting beliefs that treatment is helpful and acceptable [13,18]. However, there is a concomitant lack of certainty about the number of cases of PTSD that can be prevented through psychoeducation, the relative effectiveness of targeted versus universal prevention programs, and the optimum timing of interventions relative to traumatic incidents [19,20,21]. Broadly, there is a lack of research demonstrating the success of prevention interventions among ESW. In recent years, a call has emerged to develop and test prevention programs in well-designed longitudinal studies within affected populations [19,22].

In the absence of clear evidence-based guidelines derived from research about the best-practice in prevention, and in acknowledgment of significant delays in translating research into practice [23], affected communities have sought to develop programs for those immediately in need. An Australian example is *Behind the Seen (BTS),* a community-based, peer-led organization aiming to help reduce stigma regarding PTSD and negative mental health outcomes among emergency service workers, primarily firefighters [24]. BTS utilizes targeted, face-to-face training sessions delivered to crews and their families, covering an overview of the stresses facing first responders that can increase the risk of developing PTSD. The sessions aim to provide information on lifestyle challenges, obstacles (real and perceived) to seeking help, the meaning of stress and triggers, and available support systems.

Previous research has documented the negative impact that ESW work can have on family members, including poor mental health, strain on relationships, the reorienting of daily routines around needs, emotional distress, a loss of identity for some, and feelings of isolation [25,26,27]. Anecdotal feedback from BTS suggests that partners are motivated to understand PTSD, and how to respond appropriately to provide support for their partners, and there is therefore a need for targeted education. In recognition of these factors, BTS also delivers workshops targeting family members and friends of ESWs. Thus, the program is an ideal method for generating knowledge about potential ways to improve health promotion and prevention in affected populations.

The aim of the current study was to qualitatively examine the perceptions and experiences of participants of the Behind the Seen workshops in currently serving fire and emergency services crews. The research sought to answer the following questions: what is the perceived value of the workshops? What are the factors affecting that perceived value? Do participants report either intentions to change behavior, or actual behavior changes, as a result of attending the workshops?

## 2. Materials and Methods

### 2.1. Setting and Design

The study used a phenomenological qualitative research design to understand participants’ experiences and perceptions. Focus group methodology [28] was chosen to elicit participant experiences and to observe group interactions that contributed to shared understandings of the value of the workshops. All data collection occurred between June 2016 and February 2017, and was located in fire and rescue services across two Australian states on the East and West coasts. These services constituted participants’ place of employment or volunteering. Access to the locations was facilitated by Behind the Seen project staff.

Procedures and results are reported according to the criteria stated in the COREQ checklist [28].

### 2.2. Participants

Participants were eligible for recruitment if they had attended a BTS session and were recruited using purposive sampling techniques [29]. Each service had participated in the BTS workshop and all members were invited to participate in a focus group discussion by staff at their location. The study was promoted via invitation from BTS facilitators, who identified services that had received a BTS workshop within a given time frame (within 1, 6 or 12 months). Where services accepted the invitation, BTS facilitated introductions between the research team and the services. However, once sites were recruited, the BTS facilitators were not present during focus group discussions and did not participate in data analysis. There were no previous relationships established between the researchers and the research participants before the day of data collection. One invited participant declined to participate, owing to their employment in another emergency service and due to concerns regarding confidentiality.

### 2.3. Procedures

All data were collected by two lead facilitators (S.R.—male, A.F.—female) with PhD qualifications in public health and extensive experience conducting mixed methods research in mental health related to depression, suicide, and/or PTSD. At the time of the study, both researchers were employed at a post-doctoral level by the lead organization. The main facilitator (AF) had been previously trained in qualitatively interviewing and managing risk in populations with experiences of suicidality.

Before commencing, the lead researcher (SR) gave a general introduction to the study, including its purpose, design, collaborating partners and funding arrangements. The introduction stressed both the independence of the research team from the Behind the Seen organization and the confidentiality of data provided to the researchers. Participants were given an information sheet about the study and an opportunity to ask clarifying questions, before providing written informed consent.

Each location was attended by two or three facilitators (A.F., S.R., P.W.). The ground rules for the discussion included emphasizing confidentiality, that there were no right or wrong answers, and that the session was not intended as treatment. In total, *n* = 59 men and women participated in focus groups with between *n* = 7 and *n* = 16 participants per group. The discussions ranged between 34 and 74 min each (M = 54 min), and were digitally recorded and later transcribed for analysis.

A semi-structured interview schedule was used (see Table A1 in Appendix A) to guide the discussion, with initial “ice-breaker” questions focused on what participants had been told about the research and what they had expected upon attending. Ensuing questions focused on (i) participants’ pre-conceived ideas about BTS, (ii) their understanding of PTSD before attending the workshop, (iii) first impressions and what they remembered from the workshop, (iv) whether their thinking had changed after attending, (v) their preferred strategies for managing their mood or stress, (vi) how stress at work is generally managed in their service, and (vii) advice they would give to new recruits starting out their training. The questions were open-ended, with specified prompts used to elicit feedback regarding components of the workshop where they were not mentioned spontaneously. Though participants were not reimbursed, a light meal was provided as a token of appreciation by the research team.

Where participants disclosed any experiences with previous mental health problems or particular trauma experiences, facilitators first established whether they were happy to talk about it further in the session, then followed up at the close of the discussion to ensure they were not experiencing distress as a result. In addition, transcripts were not identified according to individual speaker, given group concerns about the privacy of both the recordings and the transcripts. The study was approved by the UNSW Human Research Ethics Committee (HREC15839).

### 2.4. Data Analysis

Iterative categorization (IC) techniques were used to code, analyze, and interpret the data. IC was chosen owing to its simplicity, transparency, and systematic rigor, which enables the coding of qualitative text by “topic, event, story, verbal interaction, signifier, feeling, idea, category, theme, concept or theory”, [30] (p1096). This technique assumes a clear relationship between the study aims, the relevant literature, and the formulation of the interview schedule to generate data, and allows for the combination of deductive coding, derived from the interview schedule, and inductive coding, derived from nascent topics arising from closer inspection of the data. IC techniques are compatible with thematic analysis [31,32], which both identifies meaningful patterns within the data [33] and acknowledges the challenges and advantages of “reflexivity” in how researchers interpret and present data [34].

For these analyses, primary coding was undertaken by AF, as derived from an initial deductive coding framework undertaken collaboratively with SR and KB. The deductive coding framework was guided by three broad questions:

What could participants remember unprompted about the BTS sessions?

What was most salient to participants, or what made the BTS sessions effective?

Did participants report doing anything differently regarding their own or other crew members’ mental health since attending the session?

Immersion in the data included re-listening to all recordings, correcting transcripts, de-identifying the data, and re-reading transcripts. Transcripts were not provided to participants and they did not comment on the findings. NVivo11 (QSR International, Melbourne, Australia) was used to manage the coding process. Following IC procedures, the initial coding sorted the data into substantial codes, placed under general headings, before descriptive analyses were used to summarize all the main points made by participants, which were then reviewed and reorganized into themes and concepts that occurred commonly, and those that were infrequent and unusual. Though the initial coding framework was deductive, the coding process and analysis allowed for the possibility of inductively identifying and interpreting themes in the data not initially accounted for. Strategies to ensure research rigor included team analysis, reflexivity, and prolonged engagement with the topic matter [35].

## 3. Results

In total, 4 major themes and 11 sub-themes were identified during analyses, which are described in greater detail below (see Table 1). The final major theme identified was not initially a key focus of the interview schedule, but was pervasive throughout the discussions, especially with regard to how participants made sense of the BTS sessions.

### 3.1. A Fertile Ground for Preventative Education about PTSD

Participants perceived their crews as “fertile ground” for preventative education about the risks of PTSD owing to (i) the relative lack of specific knowledge about PTSD, and (ii) a subsequent perceived need for initiatives such as BTS to assist in bridging the gap.

Participants described that prior to attending a BTS session, they had a poor understanding of the definition of PTSD and associated risk factors for developing PTSD. Where participants reported mental health knowledge, this mainly referred to common mental disorders such as anxiety and depression, while specific knowledge of PTSD was low.

In this context of good general knowledge about mental health, but limited understanding of the specific nature of PTSD and the links between mental health and the applied context, there was a high perceived need for the BTS sessions. This need was related to concerns about new crew members who should be aware of the risks of their work, concerns about being able to detect warning signs in others, and concerns about being able to employ early intervention with crew members thought to be showing symptoms of poor mental health. In addition, some reported that BTS was needed to convince some individuals that *all* crew members could be vulnerable to work-related mental health problems, which could in turn help to break down stigma associated with seeking help.

“If you do not have that knowledge, you’re not going to know what’s available to you, and having something like this when you start up, you might dismiss it but it’s always going to be in the back of your mind”.[Group 5]

### 3.2. Salient Aspects That Increased the Effect of the BTS Sessions

The most salient aspects of the BTS program were identified after exploring participants’ unprompted recollections of specific content from the sessions. These recollections did not differ according to the time since they had attended the session, with those who participated in a focus group immediately after the BTS session describing broadly similar features to those who had attended a BTS sessions more than 12 months ago. These salient features are described in sub-themes as follows.

#### 3.2.1. Memorable and Relatable Facilitators from Outside Their Organization

Many participants reported strong recollections of the BTS facilitators, including their demeanors, personalities, experience with fire and rescue services, and straightforward delivery styles. In particular, participants described both facilitators as relatable and authentic, with voices that resonated with their own experiences working in fire and rescue. Participants emphasized that having an ex-firefighter lead the presentation allowed them to view their personal stories as applicable to their own context, which in turn illuminated the behavior of others with whom they worked.

For many, a key feature of the presentation was the plain language used by facilitators, which allowed for simple, memorable key messages to be retained. This was further boosted by the seeming paradox of the facilitators being external to their organization but having lived experience of factors internal to their own workplaces, which enhanced the perceived trustworthiness of the information provided.

#### 3.2.2. Developing a Shared Conceptualization of PTSD, Warning Signs, and “Triggers”

After attending the BTS session, most participants felt that they understood how the causes of PTSD related to exposure to sustained stress levels and traumatic events as a result of their job. Multiple participants referenced a demonstration during the session involving glasses of water representing the different rates of stress affecting first responders relative to other workers (e.g., those in an office setting). They were less certain about causes of PTSD beyond work-related stress and were not sure whether some people were more susceptible than others.

Participants reported strong knowledge of warning signs of a mental health problem as a result of attending the BTS session, though were not always clear which warning signs were specific to PTSD. Specific recall included the following signs: irritation, unusual patterns of withdrawal or anger, increases in substance use, lack of communication, and changes in what might be considered “normal” behavior. They often described needing pre-existing knowledge of a person to be able to detect increases in risk accurately. Some described a new retrospective understanding of colleague’s behavior after hearing the facilitator talk about the lead up to their own diagnosis.

Participant 1: “Sometimes like at the station, you’d see him like go off and yell and like the anger issues, that obviously now we realise was attached to that. However, it was like you did not want to go near him because like oh, you do not know what to say, I do not know.”

Participant 2: “Additionally, then these sessions come out and the penny’s dropped”.[Group 3]

Discussing “triggers” also seemed to be a key salient feature of the presentation for most of the participants. We note there was varied use of the word trigger, with some using trigger as a stand in for warning signs in others. They described people becoming stressed and being triggered to increase alcohol consumption, or withdraw socially:

“…it might be that you’re cranky…it might be that you’re withdrawn or… go out drinking and all that; and you can see, like, you can see… people you know in stress, and what things have affected them, and you go, “It’s a trigger point.” I can see the stress affect them in that way…”.[Group 5]

In contrast, others used the word trigger specifically in relation to PTSD and to refer to previous traumatic sights, smells, or events that could generate great distress in someone with a diagnosis. This conceptualization of “triggers” was coupled with heightened awareness of the perceived need to (i) protect others in their crew from unnecessary exposures to traumatic scenes where possible, or (ii) make more conscious decisions about which scenes to attend, or roles to perform, after checking in with their own mental health while on the job. 

“If you’re aware of it you can… turn up and you can see out the front of the windscreen of the truck, I do not want to go and see that today. You’re aware of that and you do not go and do it… like, I do not want that picture in my head”.[Group 3]

Some participants described a new understanding that benign stimuli could eventually become stressful triggers as a result of repeated exposure to workplace stress. That is, for several participants, it was new information that triggers did not always refer to traumatic incidents or memories.

#### 3.2.3. Interconnectedness between Crew Mental Health and Experiences of Families and Friends

A key feature of the BTS session was the discussion between the two facilitators of the ways that workplace stress and a diagnosis of PTSD can affect family, colleagues, and friends. Many participants highlighted the strong impact of hearing both perspectives on the same problem, namely that of the person developing work-related PTSD, and that of a person observing their partner’s distress.

“From her point of view, for her to say what they went through, and you know, she’s copped just as much a hard time as he has. Additionally, it was pretty personal what they got up and did, and you’ve got to take your hat off to people who have been through a bit of grief with family. You can really appreciate it. I reckon what they did, for me personally, was really gutsy. Therefore, that made a big difference to me…I reckon that was a really good spin on it. Way better than what I’ve seen with other stuff. It was nice to have her point of view…”.[Group 2]

This was commonly mentioned as a key driver of new ways to think about mental health, risks to mental health, and the impact this has on others. Several participants reported that the BTS sessions resulted in personal realizations about the ways in which their work affected their family, particularly regarding how they handled their own stress levels. Some described a new willingness to consider talking to friends or family about their work. While most still reported a preference to talk with people with similar experiences (e.g., another crew member), the BTS facilitators’ emphasis on both perspectives was key to considering that discussions with family could be more helpful or productive than they previously believed.

In two focus groups, participants also spoke about family members attending a family session in addition to the crew’s BTS sessions. For these participants, the family sessions compounded their understanding of the interactions between crew mental health and family health. Participants reported a stronger understanding of their impact on families, as well as their family members having a stronger understanding of the pressures they faced at work and why they might behave in certain ways.

“ … family members really had to be invited to this sort of session, for them to understand how we feel…we’re not just coming back and just throwing things around the house for the sake of it after a bad accident, or bad incident, it’s for them to actually understand, oh hang on there’s something wrong with us…”.[Group 3]

#### 3.2.4. Something Can Be Done

Many perceived a general message that practical intentions and actions could be undertaken immediately following attending the session. Many participants discussed having a newfound resolve to be alert to potential warning signs of mental health problems in themselves and others, and felt more aware of the risks involved in their jobs and less prone to avoiding the issue when concerned about others. Some described regret about not following up with others sooner, and how attending BTS made them more confident to be proactive, rather than reactive. For example, one participant felt they had learned skills that could apply across nearly all aspects of their life:

“I think if we had a system like BTS, every three months for instance they do a massive day up there and teach new members from the beginning how to understand, how to read the signs, how to get help, all that, everything they teach, from the beginning. You will take it the whole way through your career and everything you do in life.”.[Group 3]

This idea was commonly paired with discussion of the props and reminders provided by BTS that could be taken home from the session (e.g., keychains, stress balls). Most participants perceived these props as both useful reminders of the content of the session, and as key tools to help initiate conversations about workplace mental health with colleagues and others outside their organization, including partners and friends. The items were framed as practical and useful tools to action after attending the session.

“That pack will help doing that, because you’ll probably just go home tonight and then wake up tomorrow and forget. However, you can be like, ‘Here, have a look at this. This is what we did’, and there’s something tangible to actually start the conversation.”.[Group 2]

One participant described keeping these reminders in easily accessible places, where they might see them before attending a job, or returning from one, so that they might function as a reminder of the session and to keep them engaged in thinking about the content.

#### 3.2.5. Story of Hope and Recovery

The BTS session’s story of hope and recovery crucially allowed room for the acceptance of aspects of PTSD risk and vulnerability. Many described the powerful effect of hearing from an ex-firefighter who had ended his career after becoming unwell yet was able to give back to his community via the BTS sessions. This message was strengthened by witnessing the facilitators interact, illustrating a partnership that had survived an experience that many previously believed would end both careers and relationships. This narrative allowed some participants to reconsider their ideas about the worth of seeking help, the need to access services sooner, and the ability of friends and family to support them through difficult experiences they found hard to articulate.

“Additionally, that’s what I think he’s so good at, and being so positive, although he relives it day to day, like I said he’s very positive, he’s very upbeat about it, because I think what spurs him on is the fact that he knows what he’s doing is helping someone”.[Group 3]

### 3.3. Changes Attributed to the Behind the Seen Experience

Multiple participants identified changes they intended to make, or had already made, reflecting the impact of the BTS session. “Changes” primarily referred to the need for open discussions at work about both the risk of developing PTSD and also critical incidents that teams attend. However, some also reported the need for discussions with friends and family members outside of the work context.

#### 3.3.1. Intentions to Change Behavior

Intentions to change behavior were largely reported by participants who had only recently attended a BTS session at the time of the focus group occurring. These participants reported intentions to take home material provided during the session and initiate discussions with family and/or friends. A few reported that they thought the packs would assist in explaining concepts they had previously had difficulty conveying to others, as well as assist friends and family to notice and identify any warning signs of declining mental health.

Less commonly, some described intentions to do things differently with regard to new recruits, including providing better education and leading and encouraging a culture of open discussion about mental health in the workplace. One participant explained that the BTS session facilitated a better understanding of their “duty of care” to new recruits, which they perceived was not currently fulfilled by existing approaches.

“I think [BTS] is a perfect example of what should be in place. Far supersedes the [organizational support] system, I think. I’ve used the [organizational support] system and it’s an utter waste of time.”.[Group 3]

In general, many attributed reframing their perspectives towards their experience during the BTS session, with intentions to act differently and to reconsider who might be a valuable source of support that they might not have considered before. For example, one participant referred to a specific exercise from the session saying:

“It’s making me realize, maybe there’s someone in those six names I’ve written that I have not really struck up a way of talking it out with them. Additionally, it could be a good option.”.[Group 1]

#### 3.3.2. Mental Health Out in the Open

Where more time had elapsed since attending the BTS session, more concrete examples were reported of a newfound openness concerning the topic of mental health in both the workplace and in their homes. This could include simple promotion of the key messages, for example, one participant reported sharing BTS material with members of another crew, another shared the same resources with family members at home. Many reported changes they had observed at their station, for example, a greater willingness to discuss stressful incidents and greater willingness to openly acknowledge when certain jobs were “rough”. Some reported changes in their own sense of responsibility to check in with themselves and with others. One participant described that since attending the BTS session, they made a point of reiterating that “everything was up for discussion”, no matter how large or small.

This was particularly important at sites where friends and family had also attended a BTS session. Some described being more honest and open with their intimate partners, which led to improved communication and feelings of being understood.

“… if you get back from a bad job, my wife she used to go, ‘why do you talk about it, I do not want to hear about dead people’. However, she did not understand that was my way of unravelling what I’ve just experienced in front of my eyes… However, after her seeing the BTS she’s gone, ‘now I understand why you do it.’ Therefore, it’s not so much of a, ‘oh why are you talking about it again’, she just sits there and listens and helps me unwind as to what’s going on”.[Group 3]

In contrast, another participant related that the sessions had allowed them to communicate to their partner that they did not want to talk immediately and needed time, space, and quiet before they were ready to discuss anything.

Finally, some participants described their involvement in proactively intervening with a crew member they were concerned about. They recounted noticing warning signs and changes in behavior, and checking in with the person, while also alerting management and family members to their concerns. This event was described as a successful application of what had been learned during the BTS session, which reinforced their confidence that they would be proactive again in the future.

Participant 1:  We had to do it.Participant 2:  We had to; he was going to fall off the edge otherwise.Participant 3:  He was going to do something stupid.Participant 1:  Additionally, I do not think we would have made the right decision, if we had not of had the awareness.Participant 2:  From BTS.Participant 1:  Correct.[Group 3]

#### 3.3.3. Unintended Consequences

There were a few unintended effects of the program, though this was uncommon. These were mainly observed by the facilitators, rather than directly communicated by participants. This included some instances where participants who now better understood their risk of PTSD became angry, or upset, about what they perceived to be lack of organizational support or action to protect ex and/or serving members of the crews. In another case, the BTS session had appeared to inadvertently increase a participant’s fear about what might happen to them in the future.

### 3.4. The Role of the Organizational Context in Mitigating the Effects of Preventative Education Initiatives

Though the interview topic guide did not specifically seek to ask about the existing organizational context, the discussions persistently conveyed how organizational cultures shaped crew behavior, the likelihood of participants accepting BTS messages, the potential to seek help in the future if needed, and their trust in the organization, both in terms of their immediate crew and their perception of the broader organization. There was a great deal of unprompted discussion about organizational “culture”, often with emphasis on the perceived “reactive” provision of services. There was discussion of services and supports participants knew were available, as well as the types of services they believed were necessary.

#### 3.4.1. Perceived Lack of Preventative Education

A key attitude expressed by multiple participants was the belief that overall, the broader organization was not doing enough regarding education, awareness, and the prevention of mental health problems. This was reported to be a problem, as it was seen to create a situation wherein mental health services were provided reactively.

There was broad agreement across all focus groups that information should be provided early to recruits, including education on their risk of developing mental health problems, and what to do in the event they found themselves struggling. Some acknowledged this information might not always be absorbed immediately, but believed it was still important to provide.

“… how are they going to get through that to get to the telephone to ask for help when they’ve never really had that guidance in the first place?”.[Group 5]

However, this attitude was not universal, with other participants noting that they had received education and phone numbers for support services at their induction, including repeated messaging about looking out for signs of distress, and this was often followed up with review exercises once their training was complete.

#### 3.4.2. Ambivalence about Accessing Services Personally

In general, participants described awareness of services that were available to them, both internally and externally. Many cited chaplaincy programs, critical incident support services (CISS), telephone support lines, and the ability to access psychologists via a general practitioner and/or employee assistance programs. However, there was considerable variation in how these services were described and the likelihood that participants would access the services. For example, one crew described CISS as highly accessible, which they viewed as a positive thing. However, they concurrently noted that in their own crew, it was preferable for more senior members to check-in with individuals before accessing the service. Others described the same service as really only useful for individuals attending particular incidents, in comparison to the sessions run by BTS, which should be available to all, regardless of work experiences. A few reiterated concerns for confidentiality and “being careful” about who you talk to in the broader organization regarding mental health and the dangers of discrimination and victimization.

Many participants outlined an unwillingness to use existing services, driven by preferences for firstly handling things by themselves, secondly, within their own crew, by talking to colleagues or their captains, or thirdly, by talking to others completely outside of the service. This could be particularly marked where individuals were in volunteer positions, rather than paid positions. In these cases, they reported they had stronger bonds with people outside their crew, who they were more likely to confide in for support.

“I’ll say it straight up, I am not one to talk, it’s just the way I was brought up, it’s sort of, your problems are your problems. You work your way through them however slowly it may be, if you need to you just take time off… I do not really put much stock into those types of situations of talking to a completely random stranger.”.[Group 4]

When pressed about what they would do if they could not solve problems alone, and would not access the services provided, the participant reported they would talk to their family first. Several reported that the BTS message of identifying peers who might support them resonated strongly with their pre-existing preferences, as they already felt they had mentors they could turn to.

Conversely, many of the same participants described a theoretical openness to the idea of accessing clinical services for mental health. They described high levels of acceptance of psychological services when the discussion was couched in terms of somebody needing help, or in terms of a theoretical crew member who might need more help than could be provided by supportive family and friends.

#### 3.4.3. Mistrust of Organizational Motives

A key sub-theme identified was the mistrust of the organizations that provide mental health services on the part of some participants. Some believed that any service provision was motivated by growing medico-legal costs, rather than genuine care for crew injured on the job. Others cited examples of senior figures promoting existing wellbeing strategies that were perceived to be undertaken too rapidly and not sufficiently informed by the experiences of those on the ground doing the job. This subsequently affected their level of trust in the services promoted to them by the broader organization. Some also described mental health-related stigma operating at higher levels in the organizations, which added to cynicism about health promotional messages to seek help for any problems.

## 4. Discussion

This study evaluated the experiences of attendees of a community-based education session regarding PTSD among ESWs. The results showed that participants experienced the sessions as a practical learning experience, which motivated them to reflect on their understandings of mental health, personal vulnerability to stress, risk of developing PTSD, and their willingness to notice changes in behavior (in themselves, or others) that could indicate a potential mental health problem. Participants’ recall of the session was strong, regardless of the time that had elapsed since they attended, with many accurately recalling specific props, exercises, and demonstrations from the session. Participants reported subsequent changes in their understanding, attitudes, and for some, behavior that they directly attributed to their attendance at the session. This included a new willingness to talk about mental health, and a willingness to consider seeking support from people they might not have previously considered. This indicates that the BTS sessions were largely successful in their stated aims of delivering education to improve knowledge of PTSD among ESW, reduce mental health-related stigma, and promote both knowledge of services and the benefits of help-seeking for mental health problems.

These positive effects of the program were due in large part to self-reported poor levels of pre-existing knowledge of PTSD, which is consistent with previous research [13], and also the perceived authenticity of the facilitators. Participants ascribed this authenticity to the facilitator’s experiences both with fire and rescue services, and with managing a chronic mental illness and its impact on their relationship. This accords with previous research, which found that contact with people with mental illness has a stronger effect on reducing stigma than education about mental illness alone [36]. The effect of the presentation was further enhanced by the BTS story of hope and partial recovery after a major career interruption, which for some, contrasted with their pre-existing beliefs about the potential harm in disclosing mental illness in the workplace. This effect of the program can potentially be understood in reference to previous work showing that reductions in stigma and changes in attitudes are greatest where contact with people with mental illnesses disputes, or “disconfirms”, prevalent negative stereotypes [37]. For many participants, the sessions were effective because they could see a concrete example of the benefits of help-seeking and disclosing mental health problems to others.

Participation in the BTS sessions, as either a crew member, or as family and friends, could lead to improvements in communication about mental health, in both the work environment and in personal relationships. This aligns with other work showing that once more open discussions about mental health occur, this can lead to greater support and understanding across multiple relationships [38,39].

Consistent with previous research, the responses of participants suggest that multiple factors operate in the participants’ context which can have a negative effect on pathways to care. Our results found strong preferences to solve problems independently [39], which can contribute to increased risk of suicide [40], strong preferences to seek support from friends, family, and trusted individuals, rather than clinical services [41], and a mistrust of available services [42] and organizational intentions, with participants citing concerns about confidentiality and “safe” people to talk to within their organizations. Similarly, we found that although there was high acceptance of accessing services by hypothetical others, there were also high levels of personal reluctance to seek help through existing organizational supports [42]. Finally, there appeared to be some significant differences in participants’ understanding and use of the word “trigger” in relation to mental health and PTSD after attending the workshop.

### Implications

BTS sessions were acceptable to attendees and are associated with gains in knowledge, changes in attitudes, and in some cases, changes in proactive behaviors to manage personal mental health and support others. Thus, the sessions appear to be a candidate for scaling up and translating to a wider audience.

A crucial feature of the presentation is the lived experience of the facilitators, echoing previous recommendations that mental health consumers be involved in all stages of mental health research and program implementation [43], owing to benefits that accrue for eventual program and service delivery [44]. Thus, a key challenge in scaling up will be identifying how to replicate the perceived authenticity of these individual facilitators, when other facilitators with different experiences are called on to deliver the material. It is unclear from the present study whether experience with fire and rescue supersedes personal experience with PTSD, or vice versa, as the critical feature that builds initial trust in the session and openness to hearing the message. Future research should examine whether both aspects are required features to maintain the impact of the presentation.

While the outcomes of the current study are largely positive, there is the potential to improve some of the course material, particularly with regard to the conceptualization of “triggers”. This may be achieved by providing a simple definition upfront, with clear indications of both the limits and inclusions of the definition. Similarly, the workshop might be improved by conveying risk factors for PTSD beyond work-related incident stress, given that many did not report strong awareness of other risk factors for developing PTSD.

Finally, there is also the outstanding challenge of measuring behavioral change in response to attendance at the sessions. While many participants self-reported changes in behavior, there was still limited reference to personally accessing formal services for mental health problems. It is not possible to be definitive about whether or not this occurred, based on the current data. Thus, future research into the impact of the BTS sessions should include consideration of baseline levels of service access, measurement of subsequent access to formal sources of support, and understanding the nature and frequency of exposure to traumatic incidents, which are likely to have a confounding effect on service use measurement.

Limitations of the current research include the inability to determine the lasting effect of the education sessions on rates of help-seeking overall. It is also a limitation that we cannot be certain all perspectives on the sessions are represented here. For each focus group discussion reported, a station captain (or equivalent), who communicated their strong support for the BTS initiative, was in attendance. It is possible that this may have affected either the group dynamics or the reported opinions. We note, however, that anonymous evaluation forms (unpublished data accessed by the research team) collected across a range of BTS sessions largely support our conclusions that the BTS session was an acceptable and effective format for the delivery of this education.

While the current study involved participants at multiple locations, in two states, across areas with varying population density, it may still be the case that stations in other areas of Australia have unique work experiences that might lead to a different reception of the BTS workshop and facilitators. Future research should determine the impact of the BTS sessions on the uptake of services and assessments of quality in different locations.

## 5. Conclusions

The current study suggests that the Behind the Seen workshops represent an acceptable delivery format, with key messaging that supports new awareness of PTSD and risks to mental health presented by occupational exposure to workplace stress and/or trauma. The learning activities in the workshop highlight preventative and help-seeking strategies that participants find engaging, memorable, and have a high likelihood of using in the future. These results have important implications for the role of ex-service organizations with lived experience, in delivering preventative strategies to reduce the burden of poor mental health among emergency service workers.

## Figures and Tables

**Table 1 ijerph-18-04418-t001:** Themes and sub-themes.

Major theme: A fertile ground for preventative education about PTSD
Major theme: Salient aspects that increased the effect of the BTS sessionsSub themes: (i) Memorable and relatable facilitators from outside the organization (ii) Developing a shared conceptualization of PTSD, warning signs, and ‘triggers’ (iii) Interconnectedness between crew mental health and experiences of family and friends (iv) Something can be done (v) Story of hope and recovery
Major theme: Changes attributed to the Behind the Seen experienceSub themes: (i) Intentions to change behaviour (ii) Mental health out in the open (iii) Unintended consequences
Major theme: The role of the organizational context in mitigating the effects of preventative education initiatives (i) Sub-themes: (ii) Perceived lack of preventative education (iii) Ambivalence about accessing services personally (iv) Mistrust of organizational methods

BTS = Behind The Seen.

## Data Availability

The data presented in this study are available on request from the corresponding author. The data are not publicly available due to the risk of identification of participants.

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
