# Peer review of "Trauma and Mental Health Awareness in Emergency Service Workers: A Qualitative Evaluation of the Behind the Seen Education Workshops"

_ijerph, 2021, doi:10.3390/ijerph18094418_

Round 1
Reviewer 1 Report
1. Title
The title is not precise, the precise title should be "Trauma-Mental Health Awareness of Emergency Service Workers: Using a Qualitative Evaluation for Behind the Seen Workshop Training"
2. Abstract
The abstract is not clear. A clear abstract should be identify the general issue, specifying the purpose of of study, conducting methods (design, study setting, sampling technique and procedure [specific time of data collection], for instance, month/year, methods of data analysis, findings and conclusion) (see page 1).
3. Introduction
Paragraph 3 is not clear, a clear should be identified by what gaps between theoretical grounding and operational of the concept.
Paragraph 4 is not clear, should be clearly indicated what and how challenges of associate trauma mental health?
Paragraph 6 is not clear, should be clearly provided and tried to to bridge the gaps of theoretical grounding and operational the concept into the study and then provided the purpose of the study. Moreover, the authors should be clearly raised the research questions. This is because research questions may guide you ask the participants, what they feel, experience, and emerging pre-conception of trauma mental health awareness.
4. Literature
Should be more added review section (heading). It will lead to identify what dimensions, issues, operational the concepts and finds the gaps of theoretical grounding. My question is, how did you operation the concept of trauma mental health awareness in Australia context? Should be clearly when using this concept and then operationalised.
5. Methods
Should be more added research design, study setting, sampling approach and procedure and interview questions section.
6. Findings
Well read
7. Discussion
Well explained
8. Conclusion
Should be added more detailed about the conclusion. It is very too short, it's not clear what main findings are advanced in theorising and practicing.
Author Response
Comment: The title is not precise, the precise title should be "Trauma-Mental Health Awareness of Emergency Service Workers: Using a Qualitative Evaluation for Behind the Seen Workshop Training"
Response: we have updated the title of the paper so that it now reads as Trauma and Mental Health Awareness in Emergency Service Workers: a qualitative evaluation of the Behind the Seen education workshops.
Comment: The abstract is not clear. A clear abstract should be identify the general issue, specifying the purpose of of study, conducting methods (design, study setting, sampling technique and procedure [specific time of data collection], for instance, month/year, methods of data analysis, findings and conclusion) (see page 1).
Response: We note that we have identified the general issue in the introductory sentences of the abstract and that we have not included subheadings in the abstract, owing to journal conventions. We have updated the months/year of data collection in the body of the paper. We have then updated the abstract as follows:
“Abstract: Emergency service workers (ESWs) are at high risk of experiencing poor mental health including posttraumatic stress disorder (PTSD). Programs led by ex-service organizations may play an unrecognized but critical role in mental health prevention and promotion. Behind the Seen (BTS) is an Australian ex-service organization that runs workshops to raise awareness and facilitate conversations around the mental health of ESWs. The purpose of the study is to conduct a qualitative evaluation of workshop participants’ experiences, to understand the acceptability and perceived usefulness over the immediate (within 1-month), intermediate (6-months) and longer-term (12-months). Participants (n=>50) were recruited using purposive sampling across five fire and rescue services in metropolitan, regional, and rural locations. Focus groups methodology was used for data collection and data were analyzed using iterative categorization techniques. Participants reported i) high perceived need for education about PTSD, ii) highly salient aspects of the presentation that made for a positive learning experience, iii) key features of changes to intentions, attitudes, and behavior, and iv) major aspects of the organizational context that affected understanding and uptake of the program’s key messages, and the importance of the lived experiences of the facilitators in the learning process. BTS was perceived as an acceptable means of delivering mental health, PTSD, and help-seeking information to ESWs. The program is a promising candidate for scaling-up and further translation.”
Comment: Introduction, Paragraph 3 is not clear, a clear should be identified by what gaps between theoretical grounding and operational of the concept. Paragraph 4 is not clear, should be clearly indicated what and how challenges of associate trauma mental health?
Response: we note the reviewer’s suggestions and have updated both paragraph 3 and paragraph 4 as follows:
“In the absence of clear evidence-based guidelines from research about best-practice in prevention, and in acknowledgment of significant delays translating research into practice, affected communities have sought to develop programs for those immediately in need. An Australian example is Behind the Seen (BTS), a community-based peer-led organization aiming to help reduce stigma regarding PTSD and negative mental health outcomes among emergency service workers, primarily firefighters [23]. BTS uti-lizes targeted, face-to-face training sessions delivered to crews and their families, covering an overview of the stresses facing first responders that can increase risk of developing PTSD. The sessions aim to provide information on lifestyle challenges, obstacles (real and perceived) to seeking help, the meaning of stress and triggers, and available support systems.
Previous research has documented the negative impact that ESW work can have on family members, including poor mental health, strain on relationships, reorienting of daily routines around their needs, emotional distress, a loss of identity for some, and feelings of isolation [24-26]. Anecdotal feedback from BTS suggests that partners are motivated to understand PTSD, how to respond appropriately to provide support for their partners, and therefore a need for targeted education. In recognition of these factors, BTS also delivers workshops targeting family members and friends of ESWs. Thus, the pro-gram is an ideal candidate for generating knowledge about potential ways to improve health promotion and prevention in affected populations.”
Comment: Paragraph 6 is not clear, should be clearly provided and tried to to bridge the gaps of theoretical grounding and operational the concept into the study and then provided the purpose of the study. Moreover, the authors should be clearly raised the research questions. This is because research questions may guide you ask the participants, what they feel, experience, and emerging pre-conception of trauma mental health awareness.
Response: we note the reviewer’s comment and have updated the final paragraph as follows:
“The aim of the current study was to qualitatively examine the perceptions and experiences of participants of the Behind the Seen workshops in currently serving fire and emergency services crews. The research sought to answer the following questions: what is the perceived value of the workshops, what are the factors affecting that perceived value, and do participants report either intentions to change behavior, or actual behavior change as a result of at-tending the workshops?
Comment: Literature, should be more added review section (heading). It will lead to identify what dimensions, issues, operational the concepts and finds the gaps of theoretical grounding. My question is, how did you operation the concept of trauma mental health awareness in Australia context? Should be clearly when using this concept and then operationalised.
Response: we note that the current review of the literature covers the affected population (firefighters), their higher incidence of PTSD and poor mental health outcomes, including experience of suicidal thoughts and rates of suicide, risk factors for poor mental health outcomes, barriers to treatments, and organisational factors affecting access to care. We have described how we have operationalised mental health literacy and psychoeducation as interventions relevant to addressing the problem. We have given Behind the Seen as an example of the type of mental health literacy and psychoeducation program that could be useful in the current context and would argue that operationalising of trauma is not required, given the broader need for general mental health literacy and psychoeducation about help-seeking and services.
Comment: Methods; should be more added research design, study setting, sampling approach and procedure and interview questions section.
Response: we note the reviewers concerns and have updated the methods sections as follows:
We have added a new paragraph:
“2.1. Setting and Design
The study used a phenomenological qualitative research design to understand participants’ experiences and perceptions. Focus group methodology was chosen to elicit participant experiences and to observe group interactions that contributed to shared understandings of the value of the workshops. All data collection occurred between June 2016 and February 2017 and was located in fire and rescue services across two Australian states. These services constituted participants’ place of employment or volunteering. Access to the locations was facilitated by Behind the Seen project staff.”
We have updated sampling and recruitment approach used in the study:
“2.2 Participants
Participants were recruited using purposive sampling techniques. Each service had participated in the BTS workshop and all members were invited to participate in a focus group discussion by staff at their location.”
We have added new paragraph to procedures:
“2.3. Procedures
All data was collected by two lead facilitators (SR - male, AF - female) with PhD qualifications in public health and extensive experience conducting mixed methods re-search in mental health related to depression, suicide, and/or PTSD. At the time of the study, both researchers were employed at a post-doctoral level by the lead organization. The main facilitator (AF) had been previously trained in qualitatively interviewing and managing risk in populations with experiences of suicidality. “
Comment: Findings; Well read
Response: we thank the reviewer for their time and expertise in reviewing our paper.
Comment: Discussion; Well explained
Response: we thank the reviewer for their time and expertise in reviewing our paper.
Comment: Conclusion; Should be added more detailed about the conclusion. It is very too short, it's not clear what main findings are advanced in theorising and practicing.
Response: In line with the Journal recommendations, we have kept the conclusion succinct, in order to convey the key finding that the workshops are an acceptable delivery format for participants, associated with changes in both behaviours and intentions and future research should consider a similar approach. The preceding section in the discussion covers the findings and the implications of those findings, while our conclusion deliberately provides a broad summary.
Reviewer 2 Report
This paper is interesting because it reports on education on PTSD. However, the experimental data are not clearly shown, and the effects and findings of training are not fully shown. Use techniques such as text mining to show quantitative numbers (frequent words, common items). For example, in "Many participants" in "3.2.1. Memorable and relatable facilitators from outside their organization Many participants reported strong recollections of the BTS facilitators, including their demeanors, personalities, experience with fire and rescue services, and straightforward delivery styles." The number of people is unknown. This is just an example, and similar sentences can be found in other places. In addition, it is necessary to clarify the extraction method of items common to the group and the criteria for judging the characteristics of the group. For example, “Specific recall included the following signs such as: irritation, unusual patterns of withdrawal or anger, increases in substance use, lack of communication, and changes in what might be considered ‘normal’ behavior ”is ambiguous. This is just an example, and similar sentences can be found in other places.Author Response
Comment: This paper is interesting because it reports on education on PTSD. However, the experimental data are not clearly shown, and the effects and findings of training are not fully shown. Use techniques such as text mining to show quantitative numbers (frequent words, common items). For example, in "Many participants" in "3.2.1. Memorable and relatable facilitators from outside their organization Many participants reported strong recollections of the BTS facilitators, including their demeanors, personalities, experience with fire and rescue services, and straightforward delivery styles." The number of people is unknown. This is just an example, and similar sentences can be found in other places. In addition, it is necessary to clarify the extraction method of items common to the group and the criteria for judging the characteristics of the group. For example, “Specific recall included the following signs such as: irritation, unusual patterns of withdrawal or anger, increases in substance use, lack of communication, and changes in what might be considered ‘normal’ behavior ”is ambiguous. This is just an example, and similar sentences can be found in other places.
Response: we note that the design of the study is not an experimental design, nor have we chosen content analysis methodologies. As such, the use of quantitative techniques such as text mining, or counting ‘number of people’ is not warranted by the current study design. As noted in the literature, counting and representation are not the primary purpose of qualitative research and there are both advantages and disadvantages to count techniques. Use of such techniques can lead to both representational and analytical over-counting where counts are conducted but do not add to the understanding of the target phenomenon (See: Maxwell 2010, Using Numbers in Qualitative Research; Sandelowski 2001, Real Qualitative Researchers Do Not Count: The Use of Numbers in Qualitative Research).
Given the qualitative design, the focus group methodology, and the stated method of data analysis being iterative categorisation, we have elected to not introduce counts at this stage or text mining.
Reviewer 3 Report
Thank you for providing me the opportunity to review this manuscript, which reports the findings of a qualitative study regarding the acceptability of a mental health training for emergency service workers. Overall, the paper has been written well, and I do not doubt that the authors are experts in the field of mental health in the context of emergency services. Hence, my concerns are not about the content of the paper but about the study methodology, and especially, its reporting. I detail my concerns below.
Abstract
Please mention the exact number of participants.
It is not clear if “the importance of the lived experiences of the facilitators in the learning process” is part of the fourth point of the results or if it is a new, fifth, point.
Please make sure that the names of the themes in the abstract, text and table, are identical.
Materials and Methods
Please specify the states (locations) where participants were recruited.
Specify inclusion and exclusion criteria.
How many participants were recruited?
How many participated?
What were reasons for not participating?
What were the sociodemographic characteristics of participants?
Please specify the relationship (if any) between BTS and the authors. For example, have authors been involved in the design or facilitation of BTS workshops? Any prior relationship?
When was the study conducted?
Workshops were in-person or via Zoom? (e.g., Covid safe)
Analysis
What was the relevant expertise of the author(s) who conducted the coding and analysis?
How did you deal with researcher bias?
From the description in the text it seems that authors conducted a light version of thematic analysis. I fully acknowledge that there are different methods of qualitative analysis, and even different approaches within ‘thematic analysis’, but it is not clear why authors opted for this ‘light’ version. It is conceivable that a more thorough thematic analysis might have yielded stronger articulated themes.
Indeed, the result section does not look very consistent and the themes do not appear to be very balanced. Some are very short, other are more elaborated and include examples. I wonder if authors could have another look and upgrade/streamline the result section.
Table 1
The layout of this table looks a bit awkward.
Overall, I recommend that authors use the COREQ criteria for reporting of the study.
I hope that these few comments may help the authors revising the otherwise interesting manuscript. Good luck.
Author Response
Thank you for providing me the opportunity to review this manuscript, which reports the findings of a qualitative study regarding the acceptability of a mental health training for emergency service workers. Overall, the paper has been written well, and I do not doubt that the authors are experts in the field of mental health in the context of emergency services. Hence, my concerns are not about the content of the paper but about the study methodology, and especially, its reporting. I detail my concerns below.
Comment: Please mention the exact number of participants.
Response: we note the reviewer’s comment and have updated the abstract with the total number of participants (n=59).
Comment: It is not clear if “the importance of the lived experiences of the facilitators in the learning process” is part of the fourth point of the results or if it is a new, fifth, point.
Response: we note that this is not a new fifth point but is rather an important aspect that we felt needed highlighting in the abstract. We have moved this point to align with the second theme (where it sits in the results section of the paper) – however we felt it needed highlighting in the abstract for those who might not read the whole paper.
Comment: Please make sure that the names of the themes in the abstract, text and table, are identical.
Response: We have ensured that the names of the themes in the text and results table are identical. Given the both the purpose of the abstract and word count limitations, we were not able to include the full major theme names in the abstract. However, we have ensured that the shortened forms in the abstract communicate what is presented in greater detail in the results section.
Comment: Materials and Methods; Please specify the states (locations) where participants were recruited.
Response: due to privacy concerns of our participants, we have not specified the exact states, but have added that the data collection and recruitment occurred on the East and West coasts of Australia.
Comment: Specify inclusion and exclusion criteria.
Response: We have now specified that participants were eligible for inclusion if they had attended a BTS workshop. There were no exclusion criteria.
Comment: How many participants were recruited? How many participated?
Response: we have now included information that n=59 participants were recruited in both the abstract and the results.
Comment: What were reasons for not participating?
Response: we have noted in the methods that only one person declined to participate, based on concerns about confidentiality related to their employment in another emergency service.
Comment: What were the sociodemographic characteristics of participants?
Response: In order to protect confidentiality of participants, we did not collect sociodemographic characteristics of the sample.
Comment: Please specify the relationship (if any) between BTS and the authors. For example, have authors been involved in the design or facilitation of BTS workshops? Any prior relationship?
Response: the researchers are not involved in the design or facilitation of the BTS workshops. We have now specified in the method section 2.2 – Participants that there was no previous relationship between researchers and participants prior to the day of data collection.
Comment: When was the study conducted?
Response: we have updated the methods section 2.1 Setting and Design to now include the dates of data collection.
Comment: Workshops were in-person or via Zoom? (e.g., Covid safe)
Response: we have now updated the methods section 2.1 Setting and Design to state that the workshops were held in person at the participants’ place of employment/volunteering.
Comment: Analysis; What was the relevant expertise of the author(s) who conducted the coding and analysis?
Response: we have now updated the method section 2.3 Procedures to include a clearer description of the expertise of the main facilitators as follows:
“All data was collected by two lead facilitators (SR - male, AF - female) with PhD qualifications in public health and extensive experience conducting mixed methods research in mental health related to depression, suicide, and/or PTSD. At the time of the study, both researchers were employed at a post-doctoral level by the lead organization. The main facilitator (AF) had been previously trained in qualitatively interviewing and managing risk in populations with experiences of suicidality.”
Comment: How did you deal with researcher bias?
Response: As noted in the paper, we used multiple strategies to ensure research rigour including immersion, team analysis, reflexivity, and prolonged engagement with the topic matter and this is consistent with reasonable practice in reporting on focus groups.
Comment: From the description in the text it seems that authors conducted a light version of thematic analysis. I fully acknowledge that there are different methods of qualitative analysis, and even different approaches within ‘thematic analysis’, but it is not clear why authors opted for this ‘light’ version. It is conceivable that a more thorough thematic analysis might have yielded stronger articulated themes.
Response: We used iterative categorisation to arrive at the broad delineation of the main themes. While it is true that a more thorough thematic analysis might have yielded different themes, the present study was not a thematic analysis. We used iterative categorisation techniques, with a predominantly deductive approach to answer three specified questions – though some allowances were made for the inclusion of inductive codes (e.g. the fourth major theme) where it was specifically relevant to the research questions and goal of the study. This study did not seek to broadly ask about, identify, and describe all major and minor themes present in the data as would be the goal or purpose in a thematic analysis.
Comment: Indeed, the result section does not look very consistent and the themes do not appear to be very balanced. Some are very short, other are more elaborated and include examples. I wonder if authors could have another look and upgrade/streamline the result section.
Response: the description of the themes represents the findings in relation to the specific research questions being answered. Some themes are short because they are straightforward and do not require further elaboration, or data analysis did not identify multiple perspectives on the theme – thus we have not included unnecessary word count. Where themes are elaborated and include either sub-themes or examples, this represents the breadth of findings or perspectives in the data.
Comment: Table 1 The layout of this table looks a bit awkward.
Response: we agree with this and have inserted a new table with a more streamlined design.
Comment: Overall, I recommend that authors use the COREQ criteria for reporting of the study.
Response: we thank the reviewer for this suggestion and have used the COREQ checklist to add multiple additions to the paper, mainly in the methods section of the paper – please see track changed document.
Round 2
Reviewer 1 Report
All comments are revised.
Author Response
Many thanks
Reviewer 2 Report
I read this paper "Maxwell 2010, Using Numbers in Qualitative Research; Sandelowski 2001, Real Qualitative Researchers Do Not Count: The Use of Numbers in Qualitative Research". Indeed, the inclusion of quantitative numbers in qualitative assessments tends to lead to wrong conclusions when the population is small. However, it does not deny the combination of quantitative evaluation and qualitative evaluation. However, even when evaluating only by qualitative evaluation, it is necessary to show the judgment criteria and logically show the result. For example, in this paper, Chapter 3.1 states, "Where participants reported mental health knowledge, this mainly referred to common mental disorders such as anxiety and depression, while specific knowledge of PTSD was low." What are the criteria for judging? (or After attending the BTS session, most participants felt that they understood how the causes of PTSD related to exposure to sustained stress levels and traumatic events as a result of their job.) At least I was reading this treatise and couldn't logically determine if the results were valid. Such sentences are occasionally found in this treatise. You need to add the results or submit the original data to complement the results.
Author Response
Response: we respectfully disagree with these suggestions. As the reviewer would be aware, qualitative research is an umbrella term covering a very broad range of epistemologies and methods of inquiry. As such, qualitative methods of data analysis and reporting are equally broad and are contested, especially according to different research paradigms available to the researcher: e.g., positivist, post-positivist, critical realism, historical realism, skeptical relativism, postmodernism, post-structuralism, critical theory, interpretivist, or social constructivist, etc. Several of this reviewer’s comments make it clear that they are interpreting the results reported in the paper from a positivist standpoint – where methodology assumes a hypothetic-deductive and verification approach. E.g. “couldn't logically determine if the results were valid” assumes validity is only assured through counting or quantifying, their initial review referred to “the experimental data” and suggested “Use techniques such as text mining to show quantitative numbers (frequent words, common items)”. These are hallmarks of a positivist research paradigm which assumes that researcher and subjects are independent of each other and subjectivity can be controlled.
We have not positioned the results in this paper from a positivist standpoint. The presented paper is informed by phenomenology and an interpretivist research paradigm; and it is presented as such. The ontology and methodologies of an interpretivist research paradigm are widely acknowledged in the extant literature as having conflicting or contested perspectives, approaches, and underlying principles to a positivist paradigm – not the least of which are strong acknowledgement that knowledge is co-produced between the researcher and participant, that truth is mediated by researcher subjectivity (not objectivity), interpretation, and meanings of ‘reality’ are negotiated both individually and through consensus. Additionally, the paper has already provided details in the method section about techniques used to ensure research rigour: immersion, prolonged engagement, team analysis, reflection on researcher ‘reflexivity’ throughout analysis (1, 2), and further, we have adhered to another reviewer’s suggestion that we report according to the COREQ criteria (even though the use of this checklist is also highly debated in the literature).
We disagree with the reviewer that we need to “add the results or submit the original data” as this ignores a fundamental dispute in the literature that has been occurring for decades. Our research team is in alignment with Braun & Clarke who assert the following:
“Although occasionally reporting percentages or frequencies is useful, in general we argue no, it not better to report percentages or frequencies. As the reason for this stance isn’t necessarily obvious, and can be a big issue, especially as those outside a qualitative paradigm often think that indicating the actual number of participants/data items or proportion of the dataset reporting a theme provides more robust evidence and validity for a qualitative analysis, we will explain our position.
There are a multitude of reasons:
- It reflects an anxiety about the validity of qualitative research practice, to some extent suggesting that somehow our analysis might not be real (i.e., it might be ‘made up’), or our themes might be anecdotal’ rather than patterned. But all research (qualitative or quantitative) relies on trust, honesty, and good research practices. Reporting actual numbers does not circumvent this issue.
- We agree with Australian health researcher Prisicilla Pyett (2003) who argued that “counting responses misses the point of qualitative research” (p. 1174), as frequency does not determine value.
- Moreover, whether something is insightful or important for answering our research questions is not necessarily determined by whether large numbers of people said it.
- Finally, because of the nature of qualitative data, we cannot assume what the absence of a certain meaning or theme in the data actually means. Consider the difference between a quantitative survey, and qualitative data collected in interview or focus group. In a quantitative survey, you ask people to select from a number of options. Reporting, and comparing, the proportions who select each of a series of response options is meaningful, because they have all been asked the same thing, and given the same response options. With an interview or focus group, the data generated from each participant can be quite different. Because, for example interviews are fluid, flexible, and interactive data collection tools, it’s not the case that every participant in an interview study discusses exactly the same issues. So if someone doing an interview study with 20 men reported that ‘12 of the men thought …’, we can’t assume that the remaining eight men didn’t think this or thought the opposite – they may have just not discussed it. So, we have no way of interpreting what is not reported in qualitative data, and this makes reporting numerical proportions somewhat deceptive and disingenuous.”
See: Should I use numbers when reporting themes?
https://cdn.auckland.ac.nz/assets/psych/about/our-research/documents/Answers%20to%20frequently%20asked%20questions%20about%20thematic%20analysis%20April%202019.pdf
The suggestions made are not suitable for the methods employed in this paper, nor would they add greater meaning, and the suggested edits would not significantly affect the results or conclusions.
- Eakin JM, Mykhalovskiy E. Reframing the evaluation of qualitative health research: reflections on a review of appraisal guidelines in the health sciences. Journal of evaluation in clinical practice. 2003;9(2):187-94.
- Korstjens I, Moser A. Series: Practical guidance to qualitative research. Part 4: Trustworthiness and publishing. European Journal of General Practice. 2018;24(1):120-4.
Reviewer 3 Report
I thank the authors for the replies and clarifications. I have no further questions regarding this manuscript.
Author Response
Many thanks